# PLAY BY THE TYPE RULES: INFERRING CONSTRAINTS FOR LLM FUNCTIONS IN DECLARATIVE PROGRAMS

## ABSTRACT

Integrating LLM powered operators in declarative query languages allows for the combination of cheap and interpretable functions with powerful, generalizable language model reasoning. However, in order to benefit from the optimized execution of a database query language like SQL, generated outputs must align with the rules enforced by both type checkers and database contents. Current approaches address this challenge with orchestrations consisting of many LLM-based post-processing calls to ensure alignment between generated outputs and database values, introducing performance bottlenecks. We perform a study on the ability of various sized open-source language models to both parse and execute functions within a query language based on SQL, showing that small language models can excel as function executors over hybrid data sources. Then, we propose an efficient solution to enforce the well-typedness of LLM functions, demonstrating 7% accuracy improvement on a multi-hop question answering dataset with 53% improvement in latency over comparable solutions.

## 1 INTRODUCTION

Language models are capable of impressive performance on tasks requiring multi-hop reasoning. In some cases, evidence of latent multi-hop logic chains have been observed with large language models (Yang et al., 2024; Lindsey et al., 2025). However, particularly with smaller language models which lack the luxury of over-parameterization, a two-step "divide-then-conquer" paradigm has shown promise (Wolfson et al., 2020; Wu et al., 2024; Li et al., 2024).

In tasks like proof verification, languages such as Lean have become increasingly popular as an intermediate representation (Moura & Ullrich, 2021). This program synthesis paradigm, or generation of an executable program to aid compositional reasoning, has been shown to improve performance on many math-based tasks (Olausson et al.; Wang et al., 2025; Xin et al., 2024). In settings requiring multi-hop reasoning over large amounts of hybrid tabular and textual data sources, the appeal of program synthesis is two-fold: not only has synthesizing intermediate representations been proven to increase performance in certain settings (Tjangnaka et al.; Shi et al., 2024; Glenn et al., 2024), but offloading logical deductions to traditional programming languages when possible allows for efficient data processing, particular in the presence of extremely large database contexts.

Existing approaches take a two-phase approach to embedding language models into typed programming languages like SQL, where a response is first generated, and an additional call to a language model is made to evaluate semantic consistency against a reference value. For example, imagine an example query with a language model function, `SELECT * FROM t WHERE city = prompt('What is the U.S. capital?')`.

A reasonable, factual generation might be "Washington D.C.". However, when integrating this output to a SQL query against a database with the "city" stored as "Washington DC", the absence of exact formatting alignment can yield unintended results that break the reasoning chains of multi-hop problems. Various approaches have been taken to solve this language model-database alignment problem: Tjangnaka et al. align unexpected LLM generations to database values by prompting gpt-3.5-turbo, and Shi et al. (2024) introduce a `check()` function to evaluate semantic consistency under a given operator (e.g. =, <, >) via few-shot prompting to a language model. By taking a post-processing approach to type alignment, these additional calls to language models introduce

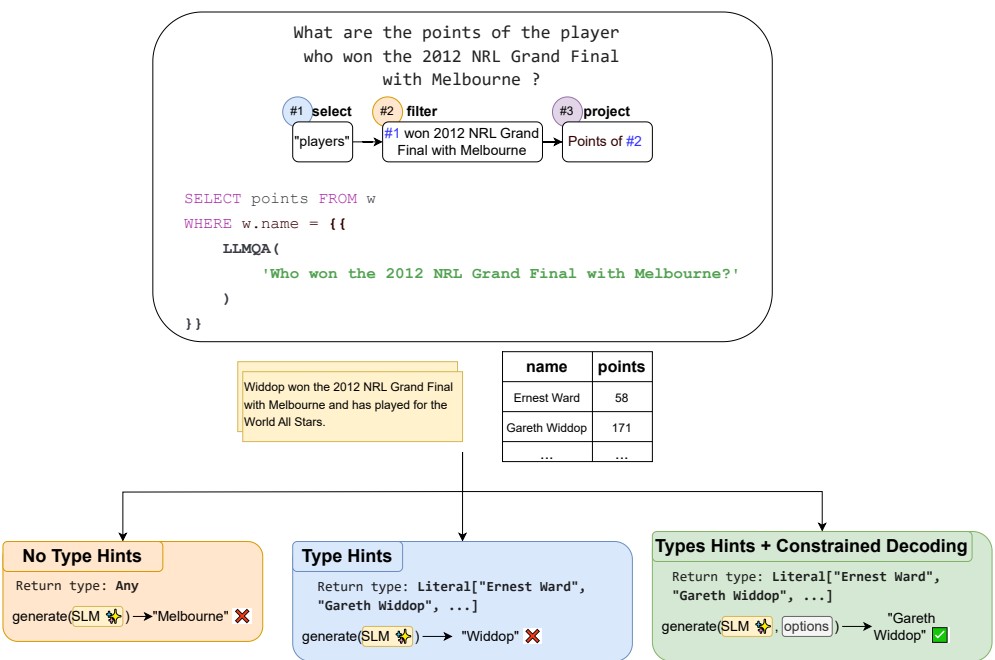

Figure 1: **Visualizing the type policies for aligning text and table values via the scalar function** LLMQA. We display the QDMR form of the question as well (Wolfson et al., 2020). Even with explicit type hints included in the prompt, small language models often fail to abide by exact formatting instructions required for precise alignment to database contexts.

bottlenecks in program execution. In performance-sensitive environments such as database systems where minimizing latency is critical, this approach is suboptimal.

Our contributions are the following:

- We propose a decoding-level type alignment algorithm for integrated LLM-DBMS systems, leveraging the type rules of SQL to infer constraints given an expression context.
- We present an efficient DB-first approach for integrating type-constrained language model functions into any database management system.
- We demonstrate the utility of small language models for generating and executing a query language for multi-hop question answering over hybrid data sources.

## 2 BACKGROUND

### 2.1 PROGRAM REPRESENTATION

We build off of BlendSQL, a query language that compiles to SQL (Glenn et al., 2024). It allows for combining deterministic SQL operators with generalizable LLM functions capable of unstructured reasoning.

Each BlendSQL function is denoted by double-curly brackets, "{{" and "}}". Using a predetermined prompt template, it generates a response from a local or remote language model with optional type-constraints to yield a function output. Given this function output, an AST transformation rule is applied to the original query AST to yield a syntactically valid SQL query, which can be executed by the native database execution engine.

Certain functions, such as the scalar LLMMAP function, rely on the creation of temporary tables to integrate function outputs into the wider SQL query. This level of integration with the DBMS allows for the scaling of BlendSQL to any database which supports the creation of temporary tables, which

expire upon session disconnect. Currently, SQLite, DuckDB, Pandas, and PostgreSQL backends are supported.

In the present work, we focus on two low-level generic functions from which complex reasoning patterns such as ranking, RAG, and entity linking can be formed.

## 2.2 LLM FUNCTIONS

For a more thorough description of the below functions, see the online documentation[1].

**LLMQA** The LLMQA function performs a reduce operation to transform a subset of data into a single scalar value. The full LLMQA prompt template can be found in Figure 4.

**LLMMAP** The LLMMAP function is a scalar function that takes a single column name and, for each value $v$ in the column, returns the output of applying $f(v)$. The full prompt LLMMAP prompt template can be found in Figure 5. We utilize prefix-caching to avoid repeated forward-passes of the prompt instruction for each of the database values, shown in Figure 2.

## 2.3 VECTOR SEARCH

All BlendSQL functions can be equipped with a FAISS (Douze et al., 2024) document store and Sentence Transformer model (Reimers & Gurevych, 2019) to perform vector search given function inputs. Additionally, BlendSQL functions may use a simplified version of the DuckDB `fmt` syntax to transfer values between subqueries, facilitating multi-hop reasoning over heterogeneous data.[2].

An example of a simple RAG workflow from the HybridQA dataset (Chen et al., 2020) is shown below.

```
SearchQA = LLMQA.from_args(
    searcher=HybridSearch(
        'all-mpnet-base-v2',
        documents=[
            "Walter Jerry Payton was an American football player...",
            "The sky is blue..."
        ],
        k=1 # Retrieve top-1 document from KNN search
    )
)
```

```
/* What is the middle name of the player with the second most National Football
    League career rushing yards ? */
SELECT {{
    LLMQA(
        'What is the middle name of {}?',
        (SELECT player FROM w ORDER BY yards DESC LIMIT 1 OFFSET 1)
    )
}}
```

## 2.4 QUERY EXECUTION

The role of a query optimizer is to determine the most efficient method for a given query to access the requested data. We implement a rule-based optimizer with a heuristic cost model. When executing a program, the query is normalized and converted to an abstract syntax tree (AST), and the nodes of the query are traversed via the standard SQL order of operations (FROM/JOIN→WHERE→GROUP BY, etc.). For each operator, the child nodes are then traversed and executed via depth-first search, with deferred execution of any LLM-based functions. In a cost planning lens, it could be said that all LLM-based functions are assigned a cost of $\infty$, whereas all native SQL operators are assigned 0. This process is visualized in Figure 2.

---

[1]To ensure anonmyity during review, documentation will be linked upon acceptance.
[2]https://duckdb.org/docs/stable/sql/functions/text.html#fmt-syntax

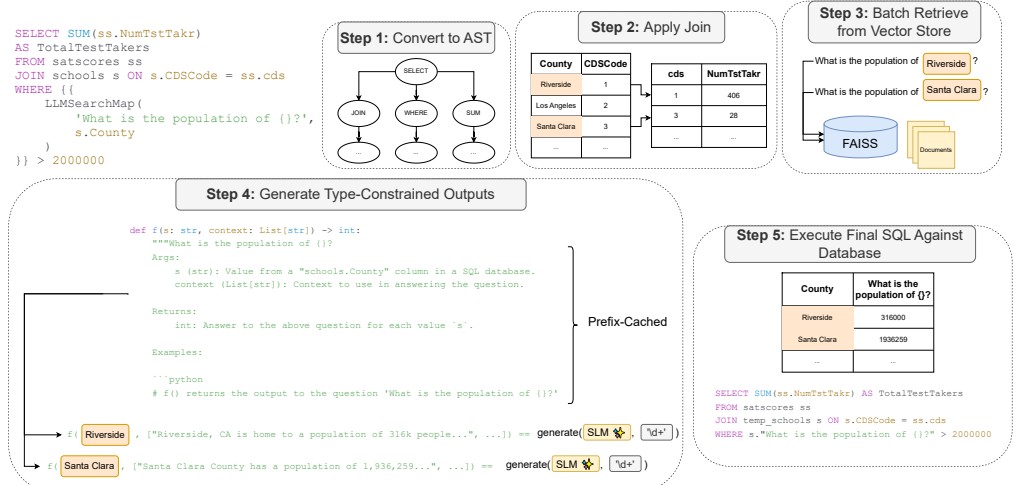

Figure 2: **Execution flow of a MAP function.** First, we apply the depth-first search described in Section 2.4 to eagerly execute the JOIN, filtering down the values required to be passed to following steps. Then, all distinct values are processed against the LLM UDF and inserted into a temporary table for usage in the final query.

Upon execution, all LLM-based functions return either a reference to a newly created temporary table or a SQL primitive, facilitating the given semantic operator it was invoked to perform. Each LLM function type is given logic to manipulate the broader query AST with this function output, denoted by TRANSFORMAST in 1. Finally, the AST is synced back to a string representation and executed against the database. With this approach, all BlendSQL queries compile to SQL in the dialect of the downstream DBMS. We define the abstract LLM function execution logic utilized in Algorithm 1 of the Appendix.

# 3 INFERRING TYPE CONSTRAINTS VIA EXPRESSION CONTEXT

When integrating LLM-based user-defined functions (UDFs) into a declarative language like SQL, it is not always clear what form the function output should take. For example, we may have the "Washington D.C." / "Washington DC" misalignment described in Section 1, as well as more explicit errors in the type checking phase of query execution.

We define three methods for handling the output of LLM UDFs below. For all methods, to accommodate our Python-style prompting patterns, we map the strings "True" and "False" to their boolean counterparts, which in turn get interpreted by "1" and "0" by SQLite. Additionally, since all database values are lowercase-normalized, we lowercase the language model output to avoid penalizing unconstrained capitalization differences. (e.g. "Washington D.C." vs. "washington d.c.").

As an illustrative example, we will take the following query. We use the aggregate function LLMQA introduced in Section 2.2, which returns a single scalar value.

```
CREATE TABLE t(
    name TEXT,
    age INTEGER
);
INSERT INTO t VALUES('Steph Curry', 37);

/* Is Lebron James older than Steph Curry? */
SELECT {{LLMQA('How old is Lebron James?')}} > age FROM t
WHERE name = 'Steph Curry'
```

### 3.1 NO TYPE HINTS

By default, the language model will be prompted to return an answer to the question with no explicit type hints or coercion. After querying the language model and applying the AST transformation rules for the aggregate LLMQA function, the final query could be:

```sql
SELECT 'The answer is 40.' > age FROM t WHERE name = 'Steph Curry'
```

Note that SQLite's type affinity allows for implicit coercion of certain TEXT literals to NUMERIC datatypes, such as the string "40". However, as an unconstrained language model with no explicit constraints may return unnecessary commentary with no applicable type conversion rules (e.g. "The answer is..."), type affinity is often rendered insufficient for executing a valid and faithful query with integrated language model output, particularly for small language models (SLMs).

### 3.2 TYPE HINTS

In this mode, a Python-style type hint is inserted into the prompt alongside the question. For the working query, this would be "Return type: int". Only the previously mentioned "True" / "False" coercion is handled by the BlendSQL interpreter, and all other outputs are inserted into the wider SQL query as a TEXT datatype.

As with the "No Type Hints" setting, type affinity rules are relied on to cast inserted language model output to most SQLite datatypes. Given the desired datatype is included via instruction in the prompt, executing with a sufficiently capable instruction-finetuned language model may yield a final SQL query of:

```sql
SELECT '40' > age FROM t WHERE name = 'Steph Curry'
```

### 3.3 TYPE HINTS & CONSTRAINED DECODING

When executed with type constraints, the process is three-fold:

1) Infer the return type of the LLM-based UDF given Table 1, and insert the Python-style type hint into the prompt.

2) Retrieve a regular expression corresponding to the inferred return type, and use it to perform constrained decoding.

3) Cast the language model output to the appropriate native Python type (e.g. INTEGER = int(s)) and perform an AST transform on the wider SQL query.

Barring any user-induced syntax errors, the output of the language model is guaranteed to result in a query that is accepted by the SQL type checker.

The resulting query in this mode would be something such as:

```sql
SELECT 40 > age FROM t WHERE name = 'Steph Curry'
```

**Database Driven Constraints** In addition to primitive types generated from pre-defined regular expressions, we also consider the LITERAL datatype as all distinct values from a column. This enables alignment between LLM generations and database contents at the decoding level, in a single generation pass. We represents these type hints by inserting "Literal['a', 'b', 'c']" in our prompts. Figure 1 demonstrates this, using an LLMQA function to align unstructured document context with a structured table.

| Function Context | Inferred Signature |
|---|---|
| `f() = TRUE` | `f() → bool` |
| `f() > 40` | `f() → int` |
| `f() BETWEEN 60.1 AND 80.3` | `f() → float` |
| `city = f()`* | `f() → Literal['Washington DC', 'San Jose']` |
| `team IN f()`* | `f() → List[Literal['Red Sox', 'Mets']]` |
| `ORDER BY f()` | `f() → Union[float, int]` |
| `SUM(f())` | `f() → Union[float, int]` |
| `SELECT * FROM VALUES f()`* | `f() → List[Any]` |

Table 1: **Sample of type inference rules for BLENDSQL UDFs.** Highlighted values indicate references to all distinct values of the predicate's column argument. Asterisks (*) refer to rules which only apply to the aggregate LLMQA function. In "Type Hints & Constrained Decoding" mode, each return type is used to fetch a regular expression to guide generation of function output (e.g. `int → \d+`).

## 4 EXPERIMENTS

### 4.1 EFFICIENCY AND EXPRESSIVITY

We first validate both the efficiency and expressivity of BlendSQL as an intermediate representation by comparing against LOTUS (Patel et al., 2024b) on the TAG-benchmark questions. LOTUS is a declarative API for data processing with LLM functions, whose syntax builds off of Pandas (pandas development team, 2020). TAG-Bench is a dataset built off of BIRD-SQL dataset (Li et al., 2023) for text-to-SQL. The annotated queries span 5 domains from BIRD, and each requires reasoning beyond what is present in the given database. For example, given the question "How many test takers are there at the school/s in a county with population over 2 million?", a language model must apply a map operation over the `County` column to derive the estimated population from either its parametric knowledge. The average size of tables in the TAG-Bench dataset is 53,631 rows, highlighting the need for efficient systems. We show the execution flow of this example in Figure 2.

Table 2 shows the sample-level latency of LOTUS and BlendSQL programs on 60 questions from the TAG-Bench dataset. Using the same quantized Llama-3.1-8b and 16GB RTX 5080, latency decreases by 53% from 1.7 to 0.76 seconds, highlighting the efficiency of BlendSQL, in addition to the expressivity of the two simple map and reduce functions. Full details of the benchmark implementations are included in Appendix B.

| Program | Model | Hardware | Execution Time (s) (↓) | Avg. Tokens per Program (↓) |
|---|---|---|---|---|
| LOTUS | Llama-3.1-70b-Instruct | 8 A100 | 3 | 127 |
| | Llama-3.1-8b-Instruct.Q4 | 1 RTX 5080 | 1.7 (+/- 0.06) | |
| BlendSQL | Llama-3.1-8b-Instruct.Q4 | 1 RTX 5080 | 0.76 (+/- 0.002) | 76 |

Table 2: **Latency measures for LLM-based data analysis programs on TAG-Bench.** For RTX 5080 results, average runtime across 5 runs is displayed. Llama-3.1-70b-Instruct results are taken from Biswal et al. (2024).

### 4.2 HYBRID QUESTION ANSWERING EXPERIMENTS

We evaluate our program synthesis with type constraints approach on the HybridQA dataset (Chen et al., 2020), containing questions requiring multi-hop reasoning over both tables and texts from Wikipedia. For example, given a question "What are the points of the player who won the 2012 NRL Grand Final with Melbourne?", a Wikipedia article must be referenced to find the winner of the NRL Grand Final, but the value of this player's points is only available in a table. While the table values contain explicit links to unstructured article, we explore a more realistic unlinked

setting, where the unstructured content must be retrieved via some RAG-like method. We evaluate our approaches on the first 1,000 examples from the HybridQA validation set. On average across the validation set, the tables have 16 rows and 4.5 columns, and the unstructured text context is 9,134 tokens.

**Metrics**  We adopt the official exact match (EM) and F1 metrics provided by the HybridQA authors, as well as semantic denotation accuracy used in Cheng et al. (2023). This denotation accuracy is more robust to structural differences between predictions and ground truth annotations pointing to the same semantic referent (e.g. "two" vs. "2").

**Models**  In order to evaluate the performance of models at various sizes, we use the Llama 3 series of models (Dubey et al., 2024). Specifically, we use Llama-3.2-1B-Instruct, Llama-3.2-3B-Instruct, Llama-3.1-8B-Instruct, and Llama-3.3-70B-Instruct. Additionally, we evaluate gemma-3-12b-it (Team et al., 2024). All models except for Llama-3.3-70B-Instruct are run on 4 24GB A10 GPUs. Llama-3.3-70B-Instruct is hosted with vLLM Kwon et al. (2023) on 4 80GB A100 GPUs.

**Few-Shot Parsing**  In the parsing phase, a language model is prompted to generate a BlendSQL query given a (`question, database`) pair. We use an abbreviated version of the BlendSQL documentation[3] alongside 4 hand-picked examples from the HybridQA train split for our prompt.

**Execution**  We execute BlendSQL queries against a local SQLite database using the LLMQA and LLMMAP functions described in Section 2. Additionally, we define a LLMSEARCHMAP function, which is a map function connected to unstructured article contexts via a hybrid BM25 / vector search. For both the LLMSEARCHMAP and LLMQA search, we use `all-mpnet-base-v2` (Song et al., 2020). All article text is split into sentences before being stored in the search index. We set the number of retrieved sentences ($k$) to 1 for the LLMSEARCHMAP function, and 10 for the LLMQA function.

For all constrained decoding functionality, we use guidance (Guidance, 2023), which traverses a token trie at decoding time to mask invalid continuations given a grammar.

**Baselines**  We also evaluate traditional end-to-end approaches to the hybrid question answering task with the Llama models. In "No Context", we prompt the model with only the question in an attempt to discern how much of the HybridQA dataset exists in the model's parametric knowledge. In "All Context", the entire table and text context is passed in the prompt. In "RAG", we use the same hybrid BM25 / mpnet retriever used in the BlendSQL functions to fetch 30 sentences from the text context. The retrieved text context and all table context are passed in the prompt with the question.

## 5 RESULTS

### 5.1 IMPACT OF TYPING POLICIES ON EXECUTION ACCURACY

Figure 3 shows the impact of the typing policies described in Section 3, with different combinations of parsing and execution models. In all settings, Type Hints + Constrained Decoding outperforms the rest of the policies. We observe a steep drop-off in performance moving from the 3b model to the 1b model as a function executor.

We see the biggest performance lift when using the 3b model to execute the functions derived from the larger 70b parameter model, where denotation accuracy rises by 6.6 points after applying type constraints. This indicates that despite occasionally failing to follow exact formatting instructions when prompted, it is still possible to efficiently extract the desired response from the model's probability distribution via constrained decoding.

---

[3]https://github.com/parkervg/blendsql/blob/4ab4aa7c7a9868ad1e61626f2398ea29e67c8c3a/docs/reference/functions.md

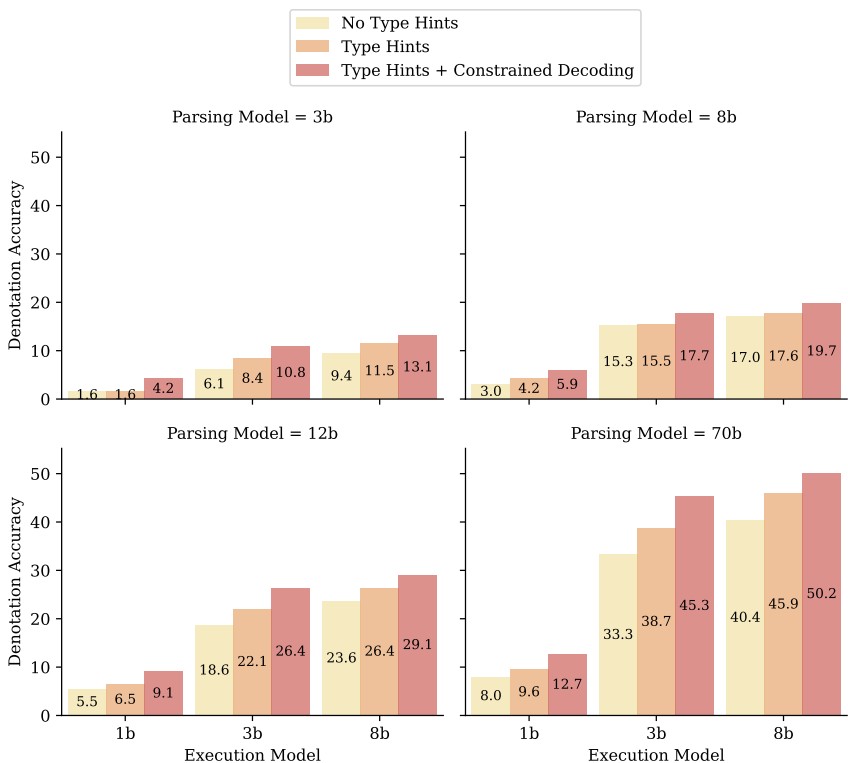

Figure 3: **Impact of various typing policies on HybridQA validation performance across model sizes.** All programs are generated using 4 few-shot examples and BlendSQL documentation. "12b" refers to gemma-3-12b-it, all other sizes refer to variants of Llama 3 Instruct. "Denotation Accuracy" refers to the semantic denotation metric used in Cheng et al. (2023). Descriptions of typing policies can be found in Section 3

## 5.2 PROGRAM SYNTHESIS VS. BASELINES

As shown in Table 3, all small models (< 70b) achieve the best performance when executing a program generated by a 70b model. The 70b model in a traditional RAG setting achieves best performance on the hybrid multi-hop reasoning dataset. Some of this success may be attributed to the model's parametric knowledge: with no context, it achieves a denotation accuracy of 6.6.

When tasked with executing a program containing the decomposition of multi-hop questions, a Llama-3.2-3b-Instruct can come close to the performance of a Llama-3.1-8b-Instruct in the RAG setting (45.3 vs. 45.6 denotation accuracy). This is notable, particularly given the fact that executed programs raised some error on 102 out of 1000 samples and fail to produce a prediction. Taking into consideration only the 899 executed programs, the 3b model achieves a denotation accuracy of 50.3. These errors are either syntax errors (e.g. missing parentheses, invalid quote escapes) or semantic errors (e.g. hallucinating a column name), and can be remedied with both rule-based post-processing or finetuning via rejection sampling. We explore the relationship between syntactic errors and downstream performance in Appendix A.1, and categorize execution errors in Table 4 .

## 6 RELATED WORK

### 6.1 COMBINING LANGUAGE MODELS WITH DATABASE SYSTEMS

Combining language models with structured data operators is a widely studied topic. To the best of our knowledge, Bae et al. (2023) were the first to propose the idea of putting calls to a neural model into a SQL query. Others have since continued exploration into domain-specific languages

| Model | Mode | Accuracy | F1 | Denotation Accuracy |
|---|---|---|---|---|
| **1b** | No Context | 1.5 | 3.85 | 2.1 |
| | All Context | 8.0 | 12.27 | 8.8 |
| | RAG | 8.9 | 12.02 | 9.7 |
| | Program Execution | **12.1** | **16.93** | **12.7** |
| **3b** | No Context | 3.1 | 5.67 | 3.6 |
| | All Context | 37.3 | 45.02 | 38.8 |
| | RAG | 35.7 | 42.57 | 37.7 |
| | Program Execution | **41.8** | **48.70** | **45.3** |
| **8b** | No Context | 3.8 | 7.38 | 4.5 |
| | All Context | 42.9 | 50.5 | 44.4 |
| | RAG | 43.8 | 50.98 | 45.6 |
| | Program Execution | **46.9** | **53.85** | **50.1** |
| **70b** | No Context | 5.7 | 10.19 | 6.6 |
| | All Context | - | - | - |
| | RAG | 54.5 | 63.55 | 57.8 |
| | Program Execution | - | - | - |

Table 3: **Results on the first 1k samples of the HybridQA validation set for Llama-Instruct models.** "Program Execution" refers to the execution of a BlendSQL program generated by Llama-3.3-70b-Instruct. Best scores for each model size are in bold.

for combining the generalized computations of language models with the structured reasoning of traditional database query languages (Cheng et al., 2023; Dorbani et al.; Tjangnaka et al.; Patel et al., 2024a).

These approaches integrate language models with database management systems at varying levels. While Patel et al. (2024a) intervenes via a Pandas API, Dorbani et al. build out a set of custom UDFs for the online analytical processing DBMS DuckDB (Raasveldt & Mühleisen, 2019). Tjangnaka et al. build out UDFs for the PostgreSQL DBMS (PostgreSQL, 2025), with additional calls to language models to determine the semantic equivalency LM-generated values against native database values.

A subset of work specifically explores efficient methods for optimizing LLM functions in relational systems (Kim et al., 2024; Liu et al., 2024).

## 6.2 CONSTRAINED DECODING

Constrained decoding refers to the process of controlling the output of language models by applying masks at the decoding level, such that generations adhere to a specific pre-determined constraint (Deutsch et al., 2019). These constraints are typically encoded via regular expressions or context-free grammars, and optimized decoding engines have emerged for deriving and applying masks (Willard & Louf, 2023; Geng et al., 2023; Park et al., 2025; Dong et al., 2024; Guidance, 2023).

Most relevant to our work is Mündler et al. (2025), who present an algorithm to enforce the well-typedness of LLM-generated TypeScript code. Whereas they tackle the problem of determining whether a partial program can be completed into a well-typed program, we explore type inference and constraints for integrating LLM outputs into a declarative query language.

## 7 CONCLUSION

In this work, we propose an efficient decoding-level approach for aligning the generated outputs of LLM UDFs with database contents. Additionally, we present evidence that small language models can excel as function executors on a complex multi-hop reasoning dataset when given appropriate constraints. This approach, while initially developed in a SQL-like language, can be extended to any typed declarative programming language.

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

## A  APPENDIX

---

**Algorithm 1** LLM Function Execution

---

**Require:** Query AST $A$, language model $L$, database $D$, function $F$
**Ensure:** $A'$: Transformed AST
1:  $T \leftarrow \text{TABLEREFS}(F)$  ▷ Gather all tables referenced in $F$
2:  **for** each $t$ in $T$ **do**
3:    **if** $t \notin D$ **then**
4:      $\text{MATERIALIZECTE}(D, A, t)$  ▷ Materialize CTE if needed
5:    **end if**
6:    **if** $\text{HASSESSIONTEMPTABLE}(D, t)$ **then**  ▷ Fetch previously written-to temp table, if exists
7:      $t \leftarrow \text{GETSESSIONTEMPTABLE}(D, t)$
8:    **end if**
9:  **end for**
10:  $R \leftarrow F(L, D, T)$  ▷ Get response from language model
11:  $A' \leftarrow \text{TRANSFORMAST}(A, R, \text{TYPE}(F))$  ▷ Transform AST, given response and function type
12:  **return** $A'$

---

### A.1  Exploring Training-Free Approaches

**Context-Free Grammar Guide**  Despite the efficiency of executing program-based solutions for question answering tasks, the implementation of a parsing step allows for potential execution errors. These execution errors may be due to syntax (e.g. subquery missing a parentheses), or semantics only noticeable at runtime (e.g. referencing a non-existent column). We design a context-free grammar to guide BlendSQL parsing at generation time to solve for many syntactic errors. The grammar is implemented via Lark (Lark), and we leverage guidance to translate the grammar into an optimized constrained decoding mask at generation time (Guidance, 2023). This grammar ensures that generated BlendSQL queries meet certain conditions, such as having balanced parentheses, and specialized functions are used in the correct context (e.g. LLMMAP must receive a quoted string and table reference as arguments[4]). However, the context-free grammar is unable to verify semantic constraints, such as ensuring that the table passed to LLMMAP exists within the current database.

Shown in Figure 7, despite the CFG preventing many syntax errors that would otherwise have occurred, the downstream denotation accuracy is not consistently improved. Specifically, smaller models that are more prone to simple synactic mistakes benefit more from the CFG guide, whereas the

---

[4]Since these aren't semantic constraints, this really only enforces that it *looks like* a table reference

```
LLMQA Prompt

Answer the question given the context, if provided.
Keep the answers as short as possible, without leading context.
For example, do not say 'The answer is 2', simply say '2'.

Question: {{question}}

Output datatype: {{return_type}}

{% if context is not none %}
Context:  {{context}}
{% endif % }

Answer:
```

Figure 4: **Prompt for the LLMQA function.**

```
LLMMAP Prompt

Complete the docstring for the provided Python function.
The output should correctly answer the question provided for each input value.
On each newline, you will follow the format of f({value}) == answer.

def f(s: str) -> bool:
    """Is an NBA team?
    Args:
        s (str): Value from the "w.team" column in a SQL database.

    Returns:
        bool: Answer to the above question for each value `s`.

    Examples:
        ```python
        # f() returns the output to the question 'Is an NBA team?'
        f("Lakers") == True
        f("Nuggets") == True
        f("Dodgers") == False
        f("Mets") == False
        ```
        """
        ...

def f(s: str) -> {{return_type}}:
    """{{question}}
    Args:
        s (str): Value from the {{table_name}}.{{column_name}} in a
            SQL database.

    Returns:
        {{return_type}}: Answer to the above question for each value `s`.

    Examples:
        ```python
        # f() returns the output to the question '{{question}}'
        f({{value}}) =
```

Figure 5: **Prompt for the LLMMAP function.** The instruction and few-shot example(s) are prefix cached, enabling quick batch inference over the sequence of database values.

large Llama-3.3-70b-Instruct is actually harmed by the constraints. We hypothesize this may be due to errors in the Lark CFG or guidance's use of fast-forward tokens[5], though leave deeper exploration of this to future work.

---

[5]https://github.com/guidance-ai/llguidance/blob/main/docs/fast_forward.md

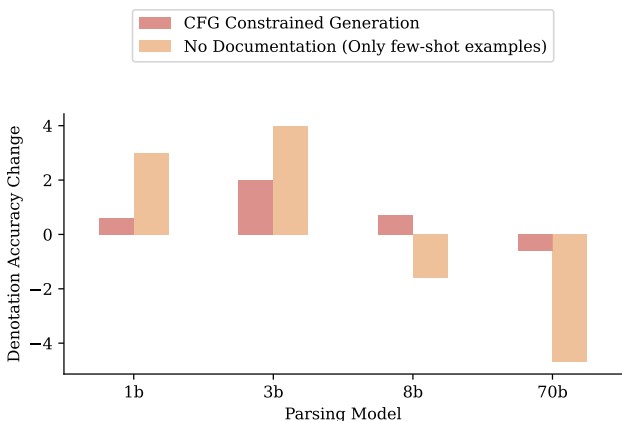

Figure 6: **Impact of ablations for different parsing models.** While larger models show decreased performance when removing descriptive documentation, smaller models exhibit moderate gains. Results shown use a Llama-3.1-8b-Instruct as a function executor in the "Type Hints + Constrained Decoding" setting, described in Section 3.3.

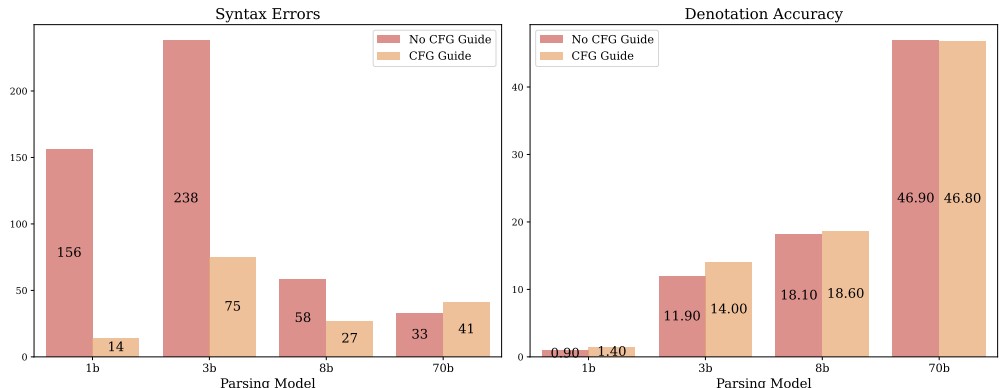

Figure 7: **Decreasing syntax errors isn't strongly correlated with improved downstream performance**. The real difficulty of semantic parsing lies in the semantic alignment, not shallow syntactic grammaticality. Results shown use a Llama-3.1-8b-Instruct as a function executor in the "Type Hints + Constrained Decoding" setting, described in Section 3.3.

## B    BENCHMARKING DETAILS

Both systems are evaluated on the same RTX 5080 16GB GPU. The max context length is set to 8000 for all evaluations.

**BlendSQL Setup**   We use blendsql==0.0.48 for our runtime experiment.  We use llama-cpp-python version 0.3.16, pointing to llama.cpp@4227c9be4268ac844921b90f31595f81236bd317. The Q4_K_M quant from bartowski/Meta-Llama-3.1-8B-Instruct-GGUF model is used.

**LOTUS Setup**   We use lotus-ai==1.1.3 for our runtime experiment.  Generation is performed using ollama version 0.6.7, which uses llama.cpp@e54d41befcc1575f4c898c5ff4ef43970cead75f as its backend. The Q4_K_M quant, referenced by ollama via llama3.1:8b, is used.

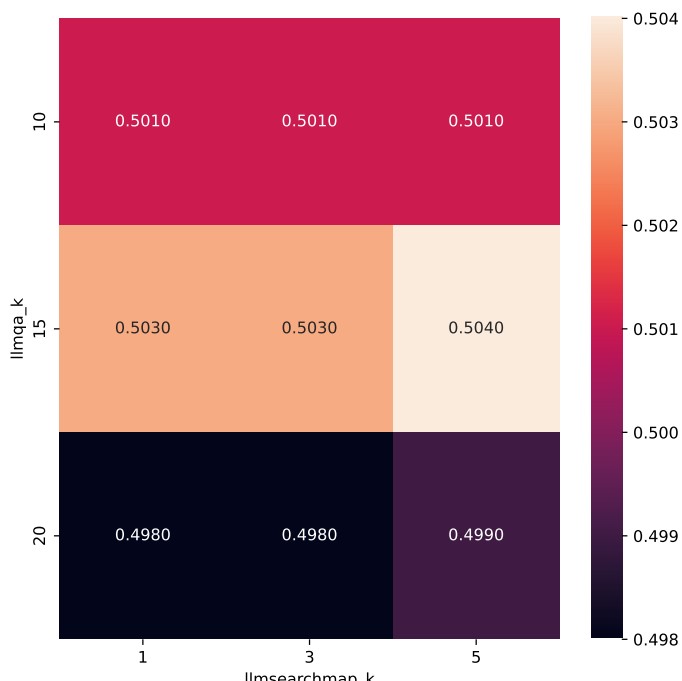

Figure 8: Hyperparameter sweeps for various settings of $k$ in our hybrid vector search components

| Error Type | Count |
|---|---|
| Empty LLMQA Context | 48 |
| Generic SQLite Syntax | 13 |
| BlendSQL Column Reference Error | 13 |
| Hallucinated Column | 11 |
| Tokenization Error | 6 |
| Hallucinated Table | 4 |
| F-String Syntax | 1 |
| Misc. | 1 |

Table 4: **Categorization of execution errors raised by programs generated by Llama-70-Instruct.** Results shown are from 1000 examples from the HybridQA validation set.

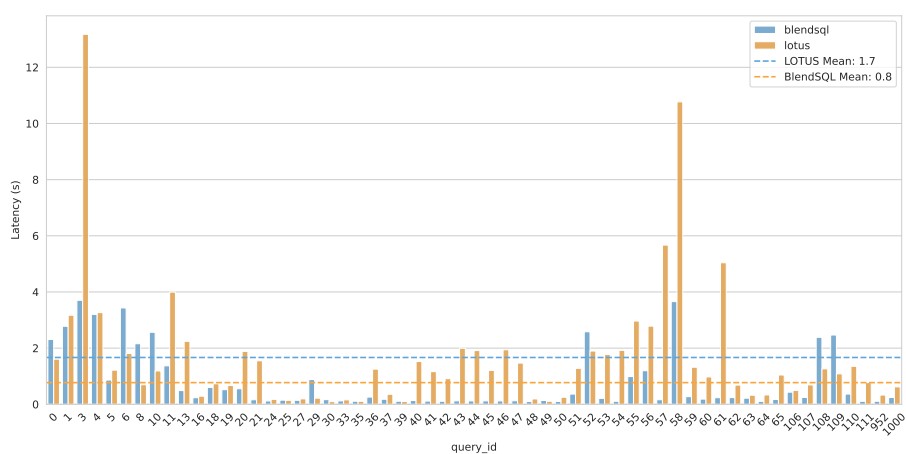

Figure 9: **Sample level latency of declarative LLM programs across question types on TAG-Benchmark.** Results shown are averaged across 5 runs on an RTX 5080.

## C  EXAMPLE PROGRAMS

Below contains a short sample of BlendSQL queries generated by Llama-3.3-70b-Instruct.

```
/* What is the difference in time between José Reliegos of Spain and the person
    born 5 September 1892 who competed at the 1928 Olympics ? */
SELECT
    CAST(REPLACE("time", ':', '.') AS REAL) -
    (SELECT CAST(REPLACE("time", ':', '.') AS REAL)
     FROM w
     WHERE athlete = {{
        LLMQA(
            'Who was born on 5 September 1892 and competed at the 1928 Olympics?'
        )
    }})
FROM w
WHERE athlete = 'josé reliegos'

/* Which # 1 ranked gymnast is the oldest ? */
WITH t AS (
    SELECT gymnasts FROM w
    WHERE rank = 1
) SELECT gymnasts FROM t
ORDER BY {{LLMSearchMap('What year was {} born?', t.gymnasts)}} ASC LIMIT 1

/* What city is the university that taught Angie Barker located in ? */
SELECT {{
    LLMQA(
        'In what city is {}?',
        (SELECT institution FROM w WHERE name = 'angie barker')
    )
}}

/* In which city is this institute located that the retired American
    professional basketball player born on November 23 , 1971 is affiliated with
    ? */
SELECT {{
    LLMQA(
        'In which city is {} located?',
        (
            SELECT "school / club team" FROM w
            WHERE player = {{
                LLMQA(
                    'What is the name of the retired American professional
                    basketball player born on November 23, 1971?'
                )
```

```
              }}
          )
      )
  }}

  /* How many players whose first names are Adam and weigh more than 77.1kg? */
  SELECT COUNT(*) FROM Player p
  WHERE p.player_name LIKE 'Adam%'
  AND p.weight > {{LLMQA('What is 77.1kg in pounds?')}}

  /* Of the 5 racetracks that hosted the most recent races, rank the locations by
      distance to the equator. */
  WITH recent_races AS (
      SELECT c.location FROM races ra
      JOIN circuits c ON c.circuitId = ra.circuitId
      ORDER BY ra.date DESC LIMIT 5
  ) SELECT * FROM VALUES {{
      LLMQA(
          'Order the locations by distance to the equator (closest -> farthest)',
          options=recent_races.location,
          quantifier='{5}'
      )
  }}
```

