# OpenReview forum: "Play by the Type Rules: Inferring Constraints for LLM Functions in Declarative Programs"
_ICLR.cc/2026/Conference — ICLR 2026 Conference Withdrawn Submission_

### Official Review · Reviewer_qLHi · 2025-10-28

**Soundness:** 2
**Presentation:** 3
**Contribution:** 2
**Rating:** 4
**Confidence:** 4

**Summary:**

This paper addresses type misalignment when LLMs generate outputs inside SQL queries (e.g., LLM says "Washington D.C." but database has "Washington DC"). Proposes inferring return type constraints from SQL context (e.g., f() > 40 → must return int) and using constrained decoding to enforce type-safety at generation time. Shows 53% latency improvement over LOTUS and demonstrates 3B-8B models can execute hybrid SQL+LLM queries when properly constrained.

**Strengths:**

1. Type inference is practical. Automatically deriving constraints from SQL expressions  is elegant - city = f() → constrain to database values, f() > 40 → constrain to integers.
2. Single-pass generation is  faster. Replacing multiple LLM calls (generate → validate → maybe retry) with one constrained generation shows real speedup: 0.76s vs 1.7s on same hardware.
3. Database value alignment at decoding level is novel. Using Literal types to constrain outputs to exact database values solves the "Washington D.C." vs "Washington DC" problem without semantic similarity matching or additional LLM calls.

**Weaknesses:**

1. Only works for open-source models. Constrained decoding requires logit access. Doesn't work with GPT-4, Claude, Gemini APIs. Paper never states this limitation explicitly. Baseline systems like LOTUS and SUQL can use closed-source models (SUQL used GPT-3.5), which may be stronger. Need head-to-head comparison.
Missing critical accuracy comparisons:

2. Missing critical accuracy comparisons:
  - No comparison vs SUQL (uses GPT-3.5 for validation) - only cited in related work
  - Table 2 only compares latency vs LOTUS, not accuracy
  - Baselines like LOTUS can use closed-source models (potentially stronger) - need head-to-head accuracy comparison
  - Don't evaluate 70B as executor despite showing 70B RAG gets 57.8% vs their best 50.1%

3. Regex constraints are fragile. Pattern \d+ only allows single number sequences. Unclear how this handles answers with multiple numbers, formatted numbers (14,000,000), ranges (35-40), or complex types (latitude/longitude pairs). No analysis of failure modes or what percentage of queries need richer type expressions.

**Questions:**

1. What's the actual cost comparison? For 1M queries: self-hosting 8B (GPU, ops, engineering) vs closed-source API calls. At what scale/accuracy requirement does your approach win?

2. How does accuracy compare vs systems that can use closed-source models? LOTUS and SUQL can use GPT-4/GPT-3.5. Your method is restricted to open-source. What's the accuracy gap? Is the latency gain worth it?

3. Can you quantify the brittleness? How often do type constraints fail or lose important information (uncertainty, ranges, etc.)? What percentage of queries need complex types that regex can't handle?

---

> ### Author Response · Authors · 2025-11-12
>
> Thank you for your review and helpful comments! We note that, upon acceptance, a full open-source repo containing our query language implementation will be made available.
>
> ## W1: Only works for open-source models.
>
> We note that many closed-source models offer structured output APIs, making the regex-constrained generation we use possible by re-casting the request as JSON generation ([OpenAI](https://openai.com/index/introducing-structured-outputs-in-the-api/), [Gemini](https://ai.google.dev/gemini-api/docs/structured-output?example=recipe)). Additionally, the actively maintained open-source library `blendsql` currently enables usage of closed-source models in the ‘Type Hints’ setting described in this work (i.e. unconstrained). Importantly, for those closed source models without structured outputs, our proposed approach still offers the benefit of prompted type hints and, most importantly, optimized execution of language model functions within a SQL context by traversing the AST and minimizing the number of calls made, described in Section 2.4.
>
> ## W2: Missing critical accuracy comparisons
>
> While the purpose of section 4.1 was to highlight the latency gains, our accuracy evaluation shows that using the smaller Llama-3.1-8B-Instruct.Q6_K with BlendSQL matches the 0.55 accuracy score reported by the TAG-Bench authors (Biswal et al.) using Llama-3.1-70b-instruct. This is notable, since it shows that Type Hints + Inferred Constraints can not only substantially decrease latency, but it allows a smaller 8b model to perform on-par with a larger 70b model.
>
> We refrain from evaluating SUQL with GPT-3.5, LOTUS with GPT-4, etc, as our primary study is to evaluate the efficacy of these systems for combining hybrid structure and unstructured data, and not the performance of individual closed models at performing a given task.
>
> ## W3: Regex constraints are fragile
>
> We would like to push back against the notion of regular expressions being fragile constraints.
>
> - We enable generation of multiple numbers via a `return_type=’List[int]’` syntax in our API, which maps to a regular expression defining a Python list of integers (e.g. `[1, 2]`).
> - The numerical range setting can be efficiently recast as a selection over many options (e.g. `(35|36|37|38|39|40)`).
> - Even complex types like latitude and longitude are possible via regular expressions (see https://stackoverflow.com/questions/3518504/regular-expression-for-matching-latitude-longitude-coordinates).
> - Regarding formatted numbers, while possible to design a regular expression, we highlight that with our proposed approach, our Python-style prompts in Figures 4 and 5 maximize the conditional probability of properly formatted python integers/floats/etc, obviating the need to generate formatted numbers.
>
> Table 4 shows a breakdown of error types, showing that most errors come from flaws in the BlendSQL query parsing stage or hallucinated syntax, and none come from the limitations of a regex-based constrained generation approach.

---

> > ### Comment · Reviewer_qLHi · 2025-11-28
> >
> > Thank you for the comment. This comment partially addresses my concerns, and I have decided to keep my score.
> >
> > 1. It would be good to explicitly discuss this limitation and validate blendsql with closed-source models.
> > 2. The paper needs to better justify why the latency gains are critical, or directly compare against stronger models. Users may prefer a slower, more accurate system.
> > 3. The issue isn't whether regex can express complex patterns, but how you know which regex to apply at query time.

---

> > > ### Author Response · Authors · 2025-11-28
> > >
> > > Thank you for your comment!
> > >
> > > ## 2. The paper needs to better justify why the latency gains are critical, or directly compare against stronger models. Users may prefer a slower, more accurate system.
> > >
> > > We believe the 53% latency improvement over a competitive baseline like LOTUS is critical in database contexts. With respect to users preferring slower, more accurate systems, we believe a relevant comment to another reviewer may be helpful, pasted below.
> > >
> > > While the purpose of section 4.1 was to highlight the latency gains, our accuracy evaluation shows that using the smaller Llama-3.1-8B-Instruct.Q6_K with BlendSQL matches the 0.55 accuracy score reported by the TAG-Bench authors (Biswal et al.) using Llama-3.1-70b-instruct. This is notable, since it shows that Type Hints + Inferred Constraints can not only substantially decrease latency, but it allows a smaller 8b model to perform on-par with a larger 70b model.
> > >
> > > We refrain from evaluating SUQL with GPT-3.5, LOTUS with GPT-4, etc, as our primary study is to evaluate the efficacy of these systems for combining hybrid structure and unstructured data, and not the performance of individual closed models at performing a given task.
> > >
> > > ## 3. The issue isn't whether regex can express complex patterns, but how you know which regex to apply at query time.
> > >
> > > The main contribution of our system is to leverage type inference to determine which regex patterns should be retrieved at inference / query time. We describe the inference process in Section 3.3 and demonstrate example inference rules in Table 1. Is there a specific complex pattern you feel would be unclear to infer under our system? We can add a section to better explain how the declared column type is also used to infer generation type - so for example, if a column referenced in a predicate is an ISO 8601 date (e.g. `2022-09-27T18:00:00`), we retrieve the pre-defined regex pattern matching this type.
> > >
> > > We appreciate your review and engagement.

---

### Official Review · Reviewer_427X · 2025-10-29

**Soundness:** 2
**Presentation:** 2
**Contribution:** 2
**Rating:** 2
**Confidence:** 4

**Summary:**

The paper studied the abilities of LLMs for function executors and proposed an efficient solution to enforce the well-typedness of LLM functions. However, the writing of this paper makes it very hard to understand.

**Strengths:**

1. Proposed an efficient, integrated system for type-constrained LLM functions. Introduces a DB-first approach coupled with decoding-level constraint application on data generation formats, ensuring that LLM outputs adhere to specific database requirements. Comparative experiments show a noticeable optimization when compared against the LOTUS model.
2. Achieved efficient integration with BlendSQL by embedding type-constrained functions directly into the database execution process, leveraging the declarative nature of the query language.

**Weaknesses:**

1. Novelty Overstatement and Incremental Contribution: The paper's claimed novelty is weak. The concept of prioritizing database execution to mitigate the high cost of LLM functions has already been discussed in existing literature concerning LLM query optimization in relational workloads[1].
2. Clarity and Organization Deficiencies:
    1. Confusing Figure References: Figure 2, explicitly titled “Execution flow of a MAP function” offers a description that is overly brief. Furthermore, the figure is inappropriately used to illustrate the execution flow for the TAG-Bench experiments in Section 4.1, despite the main function for many QA tasks being the LLMQA function. This creates a conflict between the figure's specificity and its generalized application, undermining clarity.
    2. Insufficient Methodological Detail: The execution details for applying constraints in Section 3 are not fully elaborated, specifically lacking the precise mechanism for transforming type rules into decoding constraints.
3. Incomplete Experimentation and Analysis:
    1. Lack of SOTA Comparison: Experiments lack comparison against contemporaneous program synthesis models such as SuQL[2], making it difficult to assess the competitive standing of the proposed approach.
    2. Insufficient Error Analysis: The analysis of experimental results is superficial. For instance, the authors should provide a detailed breakdown of error types in the unconstrained mode and quantify how many of these errors were eliminated by the proposed method.

[1]. Shu Liu, Asim Biswal, Audrey Cheng, Xiangxi Mo, Shiyi Cao, Joseph E Gonzalez, Ion Stoica, and Matei Zaharia. Optimizing llm queries in relational workloads. CoRR, 2024.
[2]. SuQL；Shicheng Liu Jialiang Xu Wesley Tjangnaka, Sina J Semnani Chen Jie Yu, and Monica S Lam. Suql: Conversational search over structured and unstructured data with large language models.

**Questions:**

1. The authors should provide an error-type categorization table, similar to Table 4, generated without using any constraint method. Without knowing the original distribution of error types, it is impossible to accurately determine how many low-level errors the proposed method successfully avoided.
2. Please provide a clear explanation for the missing Program Execution results for the Llama-3.3-70b-Instruct model. Specifically, is the absence of data related to deployment cost or an unresolved technical issue with applying the constraints to the largest model?
3. Has the issue of an excessively large LITERAL set been considered? If a column has millions of unique values, how does the system manage the memory and context overhead of feeding this entire set to the LLM and the constraint decoder?

---

> ### Author Response · Authors · 2025-11-12
>
> Thank you for your review and helpful comments!
>
> ## W1: Novelty Overstatement and Incremental Contribution
>
> We note that while the referenced paper ‘Optimizing LLM Queries in Relational Data Analytics Workloads’ is related, our work is inherently distinct. Liu et al. ask the following question: given a presupposed need to pass *N* rows of data to a LM, what is the most efficient way to do so? They then propose a useful method for request reordering to maximize KV cache reuse.
>
> We explore a more traditional query optimization setting - given an entire SQL query, what is the smallest possible value of *N* rows we can pass to the model, while still maintaining logical integrity? We also point out that we are the first to both 1) propose leveraging the expression context of a declarative language like SQL to infer LM generation constraints, and 2) offer a fully  functioning, generalizable system for DBMS-LM integrations, built off of temporary table creation.
>
> ## W2.1: Confusing Figure References: Figure 2, explicitly titled “Execution flow of a MAP function” offers a description that is overly brief. Furthermore, the figure is inappropriately used to illustrate the execution flow for the TAG-Bench experiments
>
> Regarding the overly brief description of Figure 2 and corresponding explanation in Section 2.4, is there a specific dimension of the approach that feels underspecified?
>
> Secondly, in what manner do you feel the figure is ‘inappropriately used to illustrate the execution flow for the TAG-Bench experiments’? The claim that ‘the main function for many QA tasks being the LLMQA function’ is true, but does not negate the importance of the map function: 21 out of the 60 TAG-Bench BlendSQL programs use the LLMMap function, including the example used in Figure 2. We do not see how this "creates a conflict between the figure's specificity and its generalized application" - any clarification would be greatly appreciated and implemented into the final manuscript. We make the full set of BlendSQL queries for the TAG-Bench dataset available at https://anonymous.4open.science/r/anonymous-tag-queries-ACBC/blendsql-tag-queries.json. We note that, upon acceptance, a full open-source implementation of our approach will be made available.
>
> ## W2.2: The execution details for applying constraints in Section 3 are not fully elaborated, specifically lacking the precise mechanism for transforming type rules into decoding constraints.
>
> Perhaps unclear on our part, Section 3.3’s statement ‘Retrieve a regular expression corresponding to the inferred return type, and use it to perform constrained decoding’ describes a heuristic mapping between expression context and pre-defined regular expressions that we use to perform decoding via the guidance library: https://github.com/guidance-ai/guidance. In practice, we traverse the SQL AST using the same approach in Section 2.4, and go through a series of conditional statements to see if any of our type rules apply. We will describe this process in more detail in our final manuscript.
>
> ## W3.1: Lack of SOTA Comparison
>
> While we attempted to benchmark SUQL in our experiments, we unfortunately found the library to not be in a functioning state (see https://github.com/stanford-oval/suql/issues/38). Additionally, we note that SUQL requires a complex orchestration of 3 separate servers to run, whereas our approach can be easily set up on a single process with 5 lines of code. Our approach can also be scaled to any DBMS that allows creation of temporary tables (DuckDB, SQLite, PostgreSQL, etc.), whereas SUQL is limited to a PostgreSQL server.
>
> ## W3.2: the authors should provide a detailed breakdown of error types in the unconstrained mode and quantify how many of these errors were eliminated by the proposed method.
>
> We note that given the structure of our main ablation study, the 6.6 denotation accuracy improvement from ‘Type Hints’ to ‘Type Hints + Constrained Decoding’ (or, 66 samples out of 1000) can be solely attributed to our contribution. The error cases follow mostly the same patterns depicted in Figure 1, which is a real example from the ablation experiment. We will include more detailed error analysis in our final manuscript to clarify this.
>
> ## Q3: Has the issue of an excessively large LITERAL set been considered?
>
> Please see our top-level comment for a response to this point.

---

### Official Review · Reviewer_6pbP · 2025-10-29

**Soundness:** 2
**Presentation:** 2
**Contribution:** 2
**Rating:** 4
**Confidence:** 3

**Summary:**

In this paper, the authors introduce a database value alignment approach for enforcing type correctness when integrating LLM-based user-defined functions into declarative query languages.
It uses the SQL expression context to determine the expected output type and restricts the model’s return value to match this type before execution.
This eliminates the need for expensive post-hoc LLM validation calls commonly used in prior systems, significantly reducing latency. The approach is implemented in BlendSQL, which supports integrating LLM functions alongside traditional SQL operators over both structured and unstructured data.
Experiments on TAG-Bench and HybridQA show a reduction in execution latency and higher accuracy.
The study further finds that small LLMs can effectively act as function executors when guided by type constraints.

**Strengths:**

S1. The approach is well-motivated and fully implemented end-to-end. The authors provide a working system rather than a conceptual prototype, and demonstrate significant execution-time savings over LOTUS in practical settings (53% latency reduction on TAG-Bench).

S2. The experiments clearly show that typing policy directly affects execution accuracy: even lightweight type constraints substantially improve denotation accuracy on HybridQA, confirming that type alignment is not just a syntactic detail but an accuracy bottleneck in LLM–DB integration.

**Weaknesses:**

W1. The provided motivating examples can be reduced to binary filter logic (True/False) that does not require literal value alignment. For example:

`Select * from w where w.name = LLMQA('who won the 2012 NRL Grand Final with Melbourne')`

Can be rewritten something like:

`Select * from w where LLMQA(f' Did {w.name} won the 2012 NRL Grand Final with Melbourne?', output: True/False)`



W2. Some of the typing benefits may already be achievable through structured output APIs or existing regex-constrained generation interfaces provided by modern LLM serving stacks, raising questions about novelty of the approach.

W3. Where a column contains many distinct value, adding literal into the prompt might be very costly?

W4. The HybridQA results in Table 3 are incomplete for the strongest model. Why didn't the author run on the entire benchmark (instead of samples) and finish all the results.

**Questions:**

Please see weakness.

---

> ### Author Response · Authors · 2025-11-12
>
> Thank you for your review and helpful comments! We appreciate the call-out that ‘type alignment is not just a syntactic detail but an accuracy bottleneck in LLM–DB integration’.
>
> ## W1: The provided motivating examples can be reduced to binary filter logic
>
> This is a good observation. While possible to re-frame some examples as binary filter logic, we emphasize that many of the latency improvements gained from our proposed approach would be negated. For example, consider a situation with 100 candidate strings from the name column, each exactly 1 token in length.
>
> In our approach using the LLMQA function, 1 forward pass would be done over the prompt containing the candidate strings, and 1 token would be generated as a response. In the proposed binary filter approach, 100 prompts would be fed to the model in a forward pass, and 100 tokens (‘yes’ or ‘no’) would be generated, aggregated, and then filtered through. The latter approach, while possible, is more expensive for both the prefill and decode stage.
>
> Additionally, the binary filter logic does not enforce the constraint that we expect exactly one name to make it past this natural language filter. If more than one candidate receives ‘yes’ predictions, we would need to implement some resolution logic to pick exactly one candidate from the subset. This is inherently handled by framing the logic as an aggregate function, rather than a binary filter / map.
>
> ## W2: Some of the typing benefits may already be achievable through structured output APIs or existing regex-constrained generation interfaces
>
> Yes, we note that many closed API providers are beginning to offer structured outputs (e.g. OpenAI, Gemini). Importantly, we do not claim constrained decoding according to some regular expression as novelty, but specifically **inferring and applying type constraints given expression context in a declarative language.** To this end, we see the increasing popularity of structured outputs as a service to be complementary to our approach - these existing services can be combined with our optimized and generalizable query execution pipeline to enable operations over many DBMS (e.g. DuckDB, SQLite, PostgreSQL, etc.) and many language model providers (Transformers, LLamaCPP, Gemini, etc.).
>
> ## W3: Where a column contains many distinct value, adding literal into the prompt might be very costly?
>
> Please see our top-level comment for a response to this point.
>
> ## W4: The HybridQA results in Table 3 are incomplete for the strongest model.
>
> We are currently running the Llama 70b model on the HybridQA validation set using the program synthesis approach, and will report the results in OpenReview when finished.

---

### Official Review · Reviewer_drWG · 2025-11-02

**Soundness:** 2
**Presentation:** 3
**Contribution:** 2
**Rating:** 2
**Confidence:** 4

**Summary:**

The paper presents the use of constrained generation as a way to ensure the LLM integration with declarative programming languages results in type consistent and correct outputs, preventing the breakdown of overall pipeline. One of the interesting observations made in the paper is that the use of smaller LM as an executor of the UDF generated by a larger LLM is quite competitive.

Overall, the paper although is interesting, fails to introduce any specific novelties. It is not claiming individual components such as BlendSQL, constrained decoding, or “small LMs as executors” as novel contributions. The primary contribution seems to be the DB-aware way they combine these pieces -- this is not strong enough contribution for consideration in ICLR (although it is an interesting empirical study).

**Strengths:**

S1. Clear integration of ideas, and nice DB integration.
S2. Excellent evidence for efficiency, and nice comparison with the use of smaller models as executors.
S3. Reasonable artifacts for reproducibility of results.

**Weaknesses:**

W1. Technical novelty is rather weak, although the overall integration/engineering seems to be quite practical.
W2. Although there are other DB-integrated systems mentioned in the paper -- SUQL, DuckDB UDF etc -- there are no empirical comparisons.
W3. The biggest issue seems to be lack of focus in terms of the contributions of the paper: BlendSQL is a prior art so its evaluation in 4.1 seems to be quite out of place, constrained decoding is also well known. The interesting part seems to be the use of small LMs as executors, but this is empirical observation with no deeper insights. So, on the whole the paper seems like a idea-integration work, which may not be suitable for ICLR.
W4. The handling of LITERAL seems suitable only for settings where the number LITERALs is fairly small. On really large databases, with millions of unique LITERALs in a column sometimes, the presented approach does not seem workable.
W5. There are many systems level challenges in the integration -- especially since the entire solution hinges of temporary table creation -- that are missing in the paper: concurrency, batching across queries, effect of prefix-cache hits on throughput, and optimizer choices (join order) under realistic workloads.

**Questions:**

Please address the comments in the previous Weaknesses section.

---

> ### Author Response · Authors · 2025-11-12
>
> Thank you for your review and helpful comments! We appreciate the comments regarding ‘clear integration of ideas’ and ‘excellent evidence for efficiency’. Complete codebases, both for our query language implementation and experiments, will be made available upon acceptance.
>
> ## W1: Technical novelty is rather weak
>
> We note that we are the first work to propose inferring type constraints given expression contexts in declarative languages, and the first to propose a general-purpose, DBMS agnostic approach to integrating language models into SQL.
>
> ## W2: Although there are other DB-integrated systems mentioned in the paper, there are no empirical comparisons
>
> We consider our latency analysis against LOTUS in Figure 4.1 as a representative analysis of our approach vs. those approaches treating alignment as a post-hoc step. We will include an additional runtime analysis against DuckDB UDFs in our final manuscript.
>
>
> ## W3: The biggest issue seems to be lack of focus in terms of the contributions of the paper: BlendSQL is a prior art so its evaluation in 4.1 seems to be quite out of place, constrained decoding is also well known.
>
> Importantly, we do not claim constrained decoding according to some regular expression as novelty, but specifically *inferring and applying type constraints given expression context in a declarative language.* To this end, we see the increasing popularity of structured outputs as a service to be complementary to our approach - these existing services can be combined with our optimized and generalizable query execution pipeline to enable operations over many DBMS (e.g. DuckDB, SQLite, PostgreSQL, etc.) and many language model providers (Transformers, LLamaCPP, Gemini, etc.).
>
> In the course of this work, we refer to BlendSQL as the novel ‘Type Hints + Constrained Decoding’ setting which we introduce, and did not exist in the original BlendSQL publication.
>
>
> ## W4: The handling of LITERAL seems suitable only for settings where the number LITERALs is fairly small.
>
> Please see our top-level comment for a response to this.
>
> ## W5: There are many systems level challenges in the integration -- especially since the entire solution hinges of temporary table creation
>
> We note that the creation of temporary tables is thread-safe and obviates any concurrency-related issues, as a lock is only placed on the specific temporary table, for a specific session. Regarding batching, Figure 5 depicts our prompt format for the LLMMap function, enabling batching over individual values. We echo the considerations of prefix-cache hits and optimizer choices, and hope to explore these topics in future work.
>
> Our goal with the present work is to move the field towards thinking more on these system level challenges by proposing a rigorous and generalizable approach to embedding language models in declarative languages. As highlighted in the introduction, existing methods have been naive, including many unnecessary LLM calls and unoptimized queries. We highlight that, as described in Section 4.1, our approach is already much more efficient than LOTUS on the TAG-Bench dataset, a competitive benchmark.

---

### Author Response · Authors · 2025-11-12
**Thank you for your reviews!**

Thank you to all reviewers for their time spent crafting in-depth feedback! We highlight that, upon acceptance, a full open-source repo containing our query language implementation will be made available.

We would like to address a couple comments made by a few reviewers here.

### Incomplete Experimental Results for Llama-70b-Instruct
While originally due to compute capacity, we are now running the Llama-70b-Instruct model in the ‘Program Synthesis’ setting on the HybridQA validation set, and will report results here when ready.

### Efficiency of applying the LITERAL constraint over a set of many values

This is a great point, and something we have considered. We explore one question from the TAG-Bench dataset as an illustrative example. The question “What is the preferred foot when attacking of the player with the most Ballon d’Or awards of all time?” is translated to the following BlendSQL program:

```sql
SELECT preferred_foot FROM Player p
JOIN Player_Attributes pa ON p.player_api_id = pa.player_api_id
WHERE player_name = {{
   LLMQA(
       "Which player has the most Ballon d'Or awards?"
   )
}}
```

The table Player contains 10,848 distinct player_name values. Since all these rows have a corresponding pair in the Player_Attributes table, no rows are filtered and all 10k distinct values are passed to the LM as a candidate option in the LLMQA function. Executing the entire query above, the **runtime on a single NVIDIA 5080 is only 0.3 seconds, averaged across 5 executions.**

The reason this is possible is the mechanism that guidance uses for constructing and applying masks. Whereas some locally constrained decoding engines like [`outlines`](https://github.com/dottxt-ai/outlines) eagerly compile a graph of all possible token steps prior to generation, our selected [`guidance`](https://github.com/guidance-ai/guidance) engine constructs masks ‘on-the-fly’, meaning that the grammar compilation overhead is minimal. For the working example, the selected generation is ‘Lionel Messi’. If no other candidate strings begin with the token ‘Li-’, then the model simply needs to generate a single token, and the constrained decoding engine is able to ‘fast-forward’ to the only viable candidate string, ‘Lionel Messi’.

We are working with the guidance team to scale up the selection to > 1 million unique values, but technically speaking, there should be minimal overhead given this approach to dynamic mask construction. Importantly, the bottleneck in our approach would not be memory required by the LM, but simply construction of masks, which is a CPU-bound process.

---

### Author Response · Authors · 2025-11-23
**Following up with Requested Experiments**

Hello,

We are following up with some requested experiments.

First, to address the benchmark against DuckDB UDFs - we are in the process of finalizing an evaluation script for all items of the TAG-Bench dataset, to ensure a holistic comparison against our proposed solution, LOTUS, and DuckDB UDFs. To illustrate why our approach is superior to simple DuckDB UDFs, consider the following query from TAG-Bench:

```
SELECT COUNT(DISTINCT s.CDSCode)
    FROM schools s
    JOIN satscores sa ON s.CDSCode = sa.cds
    WHERE sa.AvgScrMath > 560
    AND LLMMap(
        'Is this a county in the California Bay Area?',
        s.County,
        'bool'
    ) = TRUE
```

The above **executes in 1.5 seconds using our proposed DBMS-agnostic temporary table approach**, and **13.9 seconds using DuckDB UDFs**. The reason for this large gap is the inherent limitations of user-defined functions without cost hints. DuckDB pushes down the predicates and passes all 58 distinct `County` values in the `schools` table to the LLM UDF. As described in Section 2.4, our approach identifies the UDF as having a cost of ∞, and hence executes both the `JOIN` and `WHERE` predicates prior to passing the minimal subset of 19 distinct values to the language model.

Second, we present the revised Table 3 below to include evaluation of Llama-3.3-70b-Instruct in the 'Program Execution' setting. As it is still prohibitively compute-expensive to run the 'All Context' setting with the 70b model, we omit it below. We now also include performance specifically on the subset of 899 examples whose predicted BlendSQL programs executed without error, as the primary focus of the current study is accurate program execution, rather than correct program parsing. With this lens, **we see the gap between the 8b RAG setting and 8b Program Execution setting widens to a difference of 9.3 Denotation Accuracy (55.7 - 46.4)**. We find that the Llama-3.3-70b-Instruct still performs best in the traditional RAG setting, showing that our approach works best for smaller models.

| Model | Mode |        Accuracy |                        | F1 | | Denotation Accuracy | |
|:------|:-----|----------------:|-----------------------:|--------:|--------:|--------:|--------:|
| | | All 1k Examples | 899 Program Executable | All 1k Examples | 899 Program Executable | All 1k Examples | 899 Program Executable |
| **1b** | No Context |             1.5 |                    1.4 | 3.85 | 3.8 | 2.1 | 2.1 |
| | All Context |             8.0 |                    7.7 | 12.27 | 12.0 | 8.8 | 8.3 |
| | RAG |             8.9 |                    8.7 | 12.27 | 11.8 | 9.7 | 9.5 |
| | Program Execution |            12.1 |                   13.5 | 16.93 | 18.8 | 12.7 | 14.5 |
| **3b** | No Context |             3.1 |                    3.0 | 5.67 | 5.5 | 3.6 | 3.6 |
| | All Context |            37.3 |                   37.7 | 45.02 | 45.3 | 38.8 | 39.3 |
| | RAG |            35.7 |                   36.0 | 42.57 | 43.0 | 37.7 | 38.3 |
| | Program Execution |            41.8 |                   46.5 | 48.70 | 54.1 | 45.3 | 50.4 |
| **8b** | No Context |             3.8 |                    3.8 | 7.38 | 7.5 | 4.5 | 4.7 |
| | All Context |            42.9 |                   42.7 | 50.5 | 50.5 | 44.4 | 44.3 |
| | RAG |            43.8 |                   44.5 | 50.98 | 51.8 | 45.6 | 46.4 |
| | Program Execution |            46.9 |                   52.2 | 53.85 | 59.9 | 50.1 | 55.7 |
| **70b** | No Context |             5.7 |                    6.0 | 10.19 | 10.1 | 6.6 | 7.0 |
| | All Context |               - |                      - | - | - | - | - |
| | RAG |            54.5 |                   55.6 | 63.55 | 64.7 | 57.8 | 59.2 |
| | Program Execution |            48.3 |                   53.7 | 55.5 | 61.7 | 51.6 | 57.4 |

Thank you again for your reviews, and we look forward to continued discussion in this rebuttal period.

---

### Note · Authors · 2025-12-02

I have read and agree with the venue's withdrawal policy on behalf of myself and my co-authors.